# A Neuroprotective Peptide Modulates Retinal cAMP Response Element-Binding Protein (CREB), Synapsin I (SYN1), and Growth-Associated Protein 43 (GAP43) in Rats with Silicone Oil-Induced Ocular Hypertension

**DOI:** 10.3390/biom15020219

**Published:** 2025-02-03

**Authors:** Gretchen A. Johnson, Raghu R. Krishnamoorthy, Ram H. Nagaraj, Dorota L. Stankowska

**Affiliations:** 1North Texas Eye Research Institute, College of Biomedical and Translational Sciences, University of North Texas Health Science Center, Fort Worth, TX 76107, USAraghu.krishnamoorthy@unthsc.edu (R.R.K.); 2Department of Microbiology, Immunology, and Genetics, College of Biomedical and Translational Sciences, University of North Texas Health Science Center, Fort Worth, TX 76107, USA; 3Department of Pharmacology and Neuroscience, College of Biomedical and Translational Sciences, University of North Texas Health Science Center, Fort Worth, TX 76107, USA; 4Department of Ophthalmology, School of Medicine, Anschutz Medical Campus, University of Colorado, Aurora, CO 80045, USA; ram.nagaraj@cuanschutz.edu

**Keywords:** neuroprotection, peptain-1, αB-crystallin, CREB signaling, neurodegeneration, glaucoma, CPP-P1

## Abstract

This study evaluated the neuroprotective potential of peptain-1 conjugated to a cell-penetrating peptide (CPP-P1) in an ocular hypertension model of glaucoma. Brown Norway (BN) rats were subjected to intraocular pressure (IOP) elevation via intracameral injection of silicone oil (SO), with concurrent intravitreal injections of either CPP-P1 or a vehicle. Retinal cross-sections were analyzed for markers of neuroprotection, including cAMP response element-binding protein (CREB), phosphorylated CREB (p-CREB), growth-associated protein-43 (GAP43), synapsin-1 (SYN1), and superoxide dismutase 2 (SOD2). Hematoxylin and eosin staining was used to assess retinal-layer thickness. SO-treated rats exhibited significant reductions in the thickness of the inner nuclear layer (INL, 41%, *p* = 0.016), inner plexiform layer (IPL, 52%, *p* = 0.0002), and ganglion cell layer (GCL, 57%, *p* = 0.001). CPP-P1 treatment mitigated these reductions, preserving INL thickness by 32% (*p* = 0.059), IPL by 19% (*p* = 0.119), and GCL by 31% (*p* = 0.057). Increased levels of CREB (*p* = 0.17) and p-CREB (*p* = 0.04) were observed in IOP-elevated, CPP-P1-treated retinas compared to IOP-elevated, vehicle-treated retinas. Although overall GAP43 levels were low, there was a modest increase in expression within the IPL and GCL in SO- and CPP-P1-treated retinas (*p* = 0.15 and *p* = 0.09, respectively) compared to SO- and vehicle-treated retinas. SO injection reduced SYN1 expression in both IPL and GCL (*p* = 0.01), whereas CPP-P1 treatment significantly increased SYN1 levels in the IPL (*p* = 0.03) and GCL (*p* = 0.002). While SOD2 expression in the GCL was minimal across all groups, a trend toward increased expression was observed in CPP-P1-treated animals (*p* = 0.16). The SO model was replicated with SO removal after 7 days and monitored for 21 days followed by retinal flat-mount preparation to assess retinal ganglion cell (RGC) survival. A 42% loss in RGCs (*p* = 0.009) was observed in SO-injected eyes, which were reduced by approximately 37% (*p* = 0.03) with CPP-P1 treatment. These findings suggest that CPP-P1 is a promising neuroprotective agent that promotes retinal ganglion cell survival and the preservation of other retinal neurons, potentially through enhanced CREB signaling in a rat model of SO-induced ocular hypertension.

## 1. Introduction

It is estimated that nearly 112 million people worldwide will be living with glaucoma in the next 15 years [1]. Vision loss from glaucoma is currently irreversible due to progressive neurodegenerative effects that occur and the limited treatment options to prevent neurodegeneration. Damage comes from multiple facets, including an increased IOP that exerts various types of cellular and metabolic stress on the retina and pinches retinal ganglion cell (RGC) axons at the optic nerve head, leading to a progressive loss of RGCs starting in the periphery [2,3,4]. These output neurons are vital in transmitting visual information to the brain; thus, therapeutics that help preserve their function are greatly needed to sustain the quality of life of glaucoma patients.

A promising potential treatment has been designed from a well-studied family of chaperone proteins, called α-crystallins, known to activate numerous cellular cascades in response to stress to preserve cell integrity. These stress response proteins, also called heat shock proteins, are found in high concentrations in the eye lens [5,6,7,8,9,10,11,12]. Peptides derived from these proteins have also been assessed for pro-survival effects in various cell types. Alpha B-crystallin is a small heat shock protein studied for its chaperoning and anti-apoptotic effects. Peptain-1 (P1), a peptide derived from αB-crystallin, has been tested across multiple glaucoma models and found to be a potent neuroprotectant [7,8,9,11,13,14,15,16,17,18,19,20,21,22,23,24,25,26,27,28,29]. We have recently demonstrated that cell-penetrating peptide (CPP) conjugated to P1 (CPP-P1) displays improved cellular uptake and enhances neuroprotective effects in several models of RGC injury [30].

Several pathways were upregulated in a recent transcriptomic analysis of adult primary RGCs from ocular hypertensive rats treated with CPP-P1 [30]. One member in some of these pathways, cAMP response element-binding protein (CREB), is a known pro-survival transcription factor often activated following growth factor signaling [31,32,33,34,35]. In a rat model of bilateral common carotid artery occlusion, CREB was found to be decreased in ischemic rat retinas in addition to dysregulation of neurotrophic and inflammatory systems [36]. AAV-2-mediated calcium/calmodulin-dependent protein kinase II (CaMKII) expression was found to increase the phosphorylation of CREB in RGCs to protect them from excitotoxic injury and optic nerve crush (ONC) injury [37]. One study proposed that extracellular vesicles derived from Schwann cells with activated CREB signaling pathways regulate reactive gliosis in rat retinas following optic nerve crush [38]. These recent studies highlight the beneficial role of CREB signaling through its downstream effects in various animal models of glaucomatous injury.

One of the downstream effects of CREB signaling in the brain is the activation of synaptogenesis and neuroplasticity pathways [32], which were found to be upregulated in the transcriptomic analysis in our previous work [30]. Growth-associated protein 43 (GAP43) and synapsin-1 (SYN1) are commonly used as markers of synaptogenesis and neuroplasticity [39,40]. Additionally, GAP43 is used as a biochemical marker of optic nerve regeneration in zebrafish and rodents [41,42,43]. SYN1 has been used as a marker of synaptic vesicle clustering [44], and its decreased activity has been noted in neurodevelopmental disorders [45]. While the precise effect of overexpression of synaptogenic genes in the retina is not fully understood, there could be a therapeutic window for maintaining RGC axon integrity. Investigating the expression of these two specific markers in the retina will help provide insight into the type of impact CREB activation may have on neuroprotection.

In addition to investigating CREB signaling, we examined the effects of CPP-P1 on mitochondrial function by evaluating the levels of the enzyme superoxide dismutase 2 (SOD2). By quenching superoxide, SOD2 acts as the mitochondrially relevant scavenger of reactive oxygen species, and an increase in its level was observed in the aqueous humor of POAG patients [46,47]. Increased expression of SOD2 was found by transcriptomics analyses in IOP-elevated Brown Norway rats [30], but the protein expression was not previously investigated within this experimental paradigm.

The current study aimed to validate previous RNA-seq findings using an SO model of acute intraocular pressure (IOP) elevation. Specifically, it focused on confirming CPP-P1 as a potential neuroprotective agent by investigating its impact on CREB protein levels and signaling pathways. The study sought to establish that CPP-P1 enhances CREB expression and activation, providing a mechanistic basis for its neuroprotective effects.

## 2. Materials and Methods

### 2.1. Animals

All animal procedures and protocols were carried out in accordance with the policies of the Association for Research in Vision and Ophthalmology (ARVO) resolution on the use of animals in ophthalmic and vision research and approved by the Institutional Animal Care and Use Committee (IACUC) at the University of North Texas Health Science Center. Brown Norway rats were purchased from Charles River and housed in the vivarium with controlled temperature, humidity, and constant dim light, with food and water provided ad libitum.

### 2.2. Peptide

CPP-P1 (VPTLK-DRFSVNLDVKHFSPEELKVK) was obtained from Peptide 2.0, Inc. (Chantilly, VA, USA), at 95–98% purity. It was reconstituted in a vehicle (PBS), aliquoted, and stored in a 1 µg/µL stock concentration at −20 °C.

### 2.3. Silicone Oil (SO) Model

The SO injection method described earlier [48,49,50] was used to induce ocular hypertension in one eye of Brown Norway rats, where the contralateral eyes were left untreated. Briefly, the rats (retired breeder 8–10 months, male) were anesthetized using the isoflurane induction @ 4% and maintenance @ 2.5% with a 0.8–1 L oxygen level, and depth of anesthesia was ensured by the toe pinch reflex. One drop of 0.5% proparacaine hydrochloride was applied to the cornea as a topical anesthetic. An ultrafine 33-gauge needle was inserted through the cornea in the distal–temporal area about 2 mm from the limbus at a low angle to avoid contact with the iris. Once inserted, SO (Alcon Laboratories, Inc., Fort Worth, TX, USA) was slowly released until a bubble filled the anterior chamber large enough to cover the pupil (about 20 µL). After the injection, the needle was held for one minute and gradually withdrawn from the anterior chamber. After the injection, a triple antibiotic ointment (bacitracin zinc, neomycin sulfate, and polymyxin B sulfate) was applied at the injection site, and the animals were allowed to recover from the anesthesia. Rats were housed with constant dim light (90 lux) throughout the duration of the experiment to reduce diurnal fluctuations in the aqueous outflow facility.

### 2.4. Intravitreal Injections of CPP-P1 or Vehicle

Immediately following the SO injection, the IOP-elevated eye was subsequently injected with either CPP-P1 or vehicle using a Hamilton syringe. A single drop of 0.5% proparacaine hydrochloride (Alcon Laboratories, Inc., Fort Worth, TX, USA) was applied to the injection eye. An ultrafine 33-gauge needle connected to a 10 microliter Hamilton syringe was used to intravitreally inject 2 μL of either CPP-P1 (2 µg) or vehicle into the vitreous chamber by insertion of the needle through the sclera, 1 mm behind the limbus region. The injection was carried out slowly, avoiding the lens during the needle insertion. After the injection, the needle was held for 1 min and gradually withdrawn from the vitreous chamber. After the injection, a triple antibiotic (bacitracin zinc, neomycin sulfate, and polymyxin B sulfate) was applied at the injection site, and the animals were allowed to recover from the anesthesia. One week following the intravitreal injection in IOP-elevated eyes, the animals were humanely euthanized by an overdose of 120 mg/kg body weight of Fatal-Plus (pentobarbital) (Covetrus, Dublin, OH, USA) administered intraperitoneally followed by intracardial injection.

### 2.5. SO Removal Protocol and Tissue Collection Timeline

Briefly, the rats were anesthetized using the isoflurane induction @ 4% and maintenance @ 2.5% with a 0.8–1 L oxygen level and ensured by toe pinch reflex. One drop of 0.5% proparacaine hydrochloride was applied to the cornea to reduce its sensitivity. Two ultrafine 33-gauge needles were used: one as a drainage needle and the other as an irrigation needle. The drainage needle was attached to a syringe without a pump to provide a route of oil outflow. The irrigation needle was attached to a syringe with irrigation solution (PBS) to flush the oil out. To do this, two incisions were made in the temporal quadrant of the cornea at the 2- and 5-o’clock positions at the edge of the oil droplet so that the tip of the drainage needle made contact with the oil. After slowly dispensing the irrigation solution and removing the oil, the drainage needle was first withdrawn, followed by the irrigation needle, and a small air bubble was injected into the anterior chamber to maintain a normal depth. The eye was pressed closed to close the corneal incisions. After, a triple antibiotic (bacitracin zinc, neomycin sulfate, and polymyxin B sulfate) was applied at the injection site, and the animals were allowed to recover from the anesthesia.

### 2.6. IOP Measurements

IOP was measured 2–3 times a week with a TonoLab tonometer (Icare Finland Oy, Espoo, Finland) under isoflurane anesthesia within 3 min post-induction. Each IOP measurement from the device was presented as the mean of six individual measurements, and ten IOP measurements were taken for each eye.

### 2.7. Pattern Electroretinography (PERG) Measurements

Pattern electroretinography (PERG) was performed following IOP elevation to assess the function of RGCs by measuring amplitude and latency, as described previously in our lab [51]. Rats were anesthetized (confirmed by toe pinch reflex) by isoflurane induction with a SomnoFlo Electronic Vaporizer (Kent Scientific Corporation, Torrington, CT, USA) @ 2.5% isoflurane and 600 mL/min oxygen and maintenance @ 2.5% isoflurane and 300 mL/min oxygen. The rats were subsequently placed on an adaptive heated stage. PERG measurements were conducted using the Jörvec instrument (Intelligent Hearing Systems, Miami, FL, USA) and the Miami PERG system (Jörvec, Miami, FL, USA) as previously described [30].

### 2.8. Immunohistochemistry of Retina Cross-Sections

The rat eyes used for histochemical analysis did not have the SO removed before euthanasia. Rat eyes were collected, fixed in 4% PFA, and then dehydrated in 70% ethanol overnight at 4 °C, after which they underwent paraffin embedding in a TissueTek-VIP (Miles Scientific, Newark, DE, USA). A Leica RM2255 Rotary Microtome (Leica Microsystems, Buffalo Grove, IL, USA) was used to obtain 7-micron sections.

To stain the sections for the target proteins of interest, tissues were taken through deparaffinization by xylene for 5 min (two times), then a series of graded ethanol washes (100%, 95%, 80%, 70%, and 50%) for 5 min each. Next, antigen retrieval was performed by submersing slides in sodium citrate buffer and heating them to 72 °C for 20 min. Subsequently, the slides underwent blocking for 60 min and were incubated in the primary antibody mixture at 4 °C overnight in a humidified chamber. The following day, the slides were washed in PBS three times for 10 min each and then incubated in the secondary antibody mixture (1:1000 dilution) at room temperature for 2 to 3 h in the dark. The slides were then incubated for 5 min in DAPI and rinsed thrice in PBS. They were then mounted with coverslips and sealed with nail polish to prevent drying. The slides were stained for CREB (CAT#9104S, Cell Signaling Technology, Danvers, MA, USA, 1:250 dilution), p-CREB (CAT#9191S, Cell Signaling Technology, 1:250 dilution), GAP43 (CAT#335000, ThermoFisher, Ann Arbor, MI, USA, 1:250 dilution), SYN1 (CAT#515200, ThermoFisher, 1:250 dilution), and SOD2 (CAT#66474-1, ThermoFisher, 1:250 dilution). Secondary antibodies used were donkey anti-mouse Alexa 488 for CREB, donkey anti-rabbit Alexa 647 for p-CREB, donkey anti-mouse Alexa 488 for GAP43, donkey anti-rabbit Alexa 594 for SYN1, and donkey anti-mouse Alexa 488 for SOD2, all diluted 1:1000 (Life Technologies, Carlsbad, CA, USA).

### 2.9. Immunohistochemistry of Whole Retinal Flat Mounts

The rat eyes described in this section had the SO removed before euthanasia. The eyes were enucleated and fixed in 4% PFA overnight at 4 °C. Next, the retinas were carefully removed and cut at 4 positions to create quadrants for staining. The retinas were washed thrice in PBS and underwent a 5 min permeabilization step before blocking for 60 min with blocking buffer (5% normal donkey serum and 5% BSA in PBS). The retinas were incubated in a primary antibody for 3 days at 4 °C, washed thrice with PBS, and incubated in a secondary antibody for 1 h at room temperature. The retinas were then mounted with coverslips and sealed with nail polish to prevent drying. The slides were stored in the dark at 4 °C after staining for RBPMS (GTX118619, GeneTex, Irvine, CA, USA, 1:200 dilution) and secondary antibody, donkey anti-rabbit Alexa 647 (1:1k dilution, Life Technologies, Carlsbad, CA, USA). Images were taken using a fluorescent BZ-X series Keyence microscope (Keyence Corporation of America, Itasca, IL, USA) and analyzed using ImageJ/Fiji (v1.54h, National Institutes of Health, Bethesda, MD, USA).

### 2.10. Hematoxylin and Eosin Staining

The rat eyes used here did not have the SO removed before euthanasia. The slides were heated to 60 °C for 1 h before being washed three times with xylene, each for 5 min. This was followed by washes in 100% ethanol twice for 2 min each, 95% ethanol for 2 min, and water once for 2 min. Next, they underwent hematoxylin (72511, ThermoFisher Scientific, Waltham, MA, USA) staining for 3 min, a water wash for 2 min, a differentiator (mild acid) for 1 min, another water wash for 2 min, Bluing (Bluing Reagent, 7301, ThermoFisher Scientific, Waltham, MA, USA) for 1 min, a water wash for 2 min, 95% ethanol for 1 min, and eosin-Y staining for 30 s. After eosin-Y, the slides were washed in 95% ethanol for 5 min, then twice in 100% ethanol for 5 min each, and then two more washes in xylene for 15 min each.

### 2.11. Statistical Analysis

Statistical analyses were performed using GraphPad Prism 10 (GraphPad Software, La Jolla, CA, USA). Unpaired *t*-tests were performed to assess differences between each group in IHC analysis; one-way ANOVA was performed for IOP data, cell counts, H&E data, and PERG data. Data were presented as mean ± standard error of the mean (SEM). A *p*-value of less than 0.05 was considered statistically different (* *p* < 0.05, ** *p* < 0.01, *** *p* < 0.001, **** *p* < 0.0001).

## 3. Results

### 3.1. Intraocular Pressure (IOP)

Measuring IOP in the SO model is crucial for understanding the extent of ocular hypertension induced by the SO, which directly impacts the retinal changes observed; thus, we first assessed the IOP elevation to establish the severity of the induced condition before analyzing retinal outcomes. Our previous studies have shown that intravitreally administered CPP-P1 does not influence intraocular pressure [22,26]. In the current study, the average baseline IOP was 20.97 mmHg. Following SO injection, IOP increased significantly to an average of 30.47 mmHg (*p* < 0.01). The data are presented as the change in IOP from baseline (Figure 1A).

### 3.2. Pattern Electroretinography (PERG)

To assess the effects of silicone oil (SO)-induced hypertension on retinal function, we measured pattern electroretinogram (PERG) responses in treated animals. The average baseline amplitude was 11.94 µV ± 0.57, and the latency was 60.13 ms ± 0.34, consistent with values previously reported in healthy eyes. Initial attempts to measure PERG responses at 7 days post-SO injection were unsuccessful due to optical interference caused by SO in the anterior chamber, which distorted the light stimulation and signal detection (Figure 1B,C). To overcome this limitation, SO was removed prior to conducting PERG measurements in a second set of experiments, enabling more accurate assessments of retinal function.

### 3.3. CPP-P1 Prevented the Loss of Cells in the GCL

To assess the neuroprotective effects of CPP-P1 on RGCs in the first set of experiments, without SO removal, we quantified the cell density in the ganglion cell layer (GCL) by counting cells in retina cross-sections spanning from the ora serrata across the retina and calculating the number per millimeter of the retina section across different treatment groups. The average number of cells in the ganglion cell layer (GCL) per millimeter was 56.8 ± 2.13 in the naïve group, 27.5 ± 1.60 in the SO + vehicle group (*p* < 0.0001), and 43.7 ± 2.00 in the SO + CPP-P1 group (*p* = 0.0002) (Figure 1D). In the SO + vehicle group, IOP elevation induced by the SO model resulted in a significant 51% reduction in GCL cell density compared to the naïve group. However, treatment with CPP-P1 markedly mitigated this cell loss, reducing the decline by approximately 29%. These findings demonstrate that CPP-P1 exerts significant neuroprotective effects in the GCL, effectively mitigating retinal ganglion cell loss induced by elevated IOP following SO injection. By preserving a substantial proportion of retinal ganglion cells, CPP-P1 highlights its potential as a therapeutic agent for protecting retinal integrity under conditions of ocular hypertension.

### 3.4. Retinal-Layer Thickness

To evaluate the structural integrity of the retina, we measured the thickness of various retinal layers, including the outer nuclear layer (ONL), inner nuclear layer (INL), inner plexiform layer (IPL), ganglion cell layer (GCL), and overall retinal thickness. The retinal layers measured were reported in microns. In the naïve group, the average thicknesses were 36.4 ± 2.21 µm for ONL, 24.1 ± 1.77 µm for INL, 35.3 ± 2.53 µm for IPL, 28.8 ± 2.35 µm for GCL, and 127.5 ± 8.70 µm for overall retinal thickness (Figure 2). In contrast, the vehicle-treated group displayed significant reductions, with averages of 32.5 ± 2.02 µm for ONL, 14.3 ± 1.60 µm for INL, 17.0 ± 2.10 µm for IPL, 12.4 ± 1.55 µm for GCL, and 82.4 ± 8.34 µm for overall thickness, reflecting losses of 11% (*p* = 0.408), 41% (*p* = 0.016), 52% (*p* = 0.0002), 57% (*p* = 0.001), and 65% (*p* = 0.014), respectively, compared to the naïve group (Figure 2A–E, Table 1). In the CPP-P1-treated group, the average thicknesses were 38.4 ± 2.08 µm for ONL, 21.9 ± 2.38 µm for INL, 23.8 ± 2.07 µm for IPL, 21.3 ± 3.12 µm for GCL, and 113.6 ± 11.09 µm for overall retinal thickness. Compared to the naïve group, this corresponded to losses of 0% for ONL, 9% for INL, 26% for IPL, and 11% for overall thickness (Table 1). CPP-P1 treatment demonstrated appreciable preservation of retinal-layer thickness compared to the vehicle group, with the following reductions in layer thinning: ONL (*p* = 0.159), INL (*p* = 0.059), IPL (*p* = 0.119), GCL (*p* = 0.057), and overall retinal thickness (*p* = 0.090). These results suggest that CPP-P1 offers significant protective effects against retinal thinning in this model.

### 3.5. CREB and Phospho-CREB Analysis

Given the critical role of CREB in mediating neuroprotective and survival pathways within retinal neurons, we assessed its expression levels in the ganglion cell layer (GCL) across different treatment groups. CREB expression in the GCL was quantified by measuring the integrated density using ImageJ. Both the naïve and SO + vehicle groups exhibited similarly low levels of CREB expression (Figure 3A). However, in the SO + CPP-P1 group, CREB expression was higher than in the naïve group, approaching statistical significance (*p* = 0.09). Additionally, the naïve group displayed approximately twice the phosphorylated CREB (p-CREB) expression relative to CREB (Table 2). In contrast, the SO + vehicle group demonstrated a reduced p-CREB-to-CREB ratio, with p-CREB levels approximately half of those observed in the naïve group (*p* = 0.12). Notably, the SO + CPP-P1 group maintained p-CREB expression similar to the naïve group (*p* = 0.86) (Figure 3B). These findings suggest that CPP-P1 treatment may enhance CREB activation, potentially contributing to its neuroprotective effects in the GCL under conditions of SO-induced stress.

### 3.6. Expression of GAP43 and SYN1s Following IOP Elevation and Treatment with CPP-P1

GAP43 is a critical marker of neuronal growth and synaptic plasticity, making its analysis essential for understanding the neuroprotective effects of CPP-P1 treatment in retinal neurons under stress conditions. GAP43 expression was quantified by measuring integrated density in the IPL and GCL. No significant differences were observed in GAP43 expression between the naïve and IOP-elevated, vehicle-treated groups. However, trends of increased GAP43 expression were noted in the IPL and GCL of CPP-P1-treated rats post IOP elevation vs. vehicle-treated controls (*p* = 0.15, *p* = 0.09, respectively) (Figure 4).

To gain a more comprehensive understanding of the neuroprotective effects of CPP-P1, we extended our analysis to include SYN1, a crucial protein in synaptogenesis and synaptic function, in addition to GAP43. SYN1 expression was measured by integrated density in the IPL and the GCL. There was a notable expression of SYN1 in the naïve IPL and a much lower expression in the GCL compared to GAP43. This expression significantly declined in the vehicle-treated group in the IPL (*p* = 0.01), and a similar trend of decreased SYN1 expression was observed in the GCL (*p* = 0.21). Rats treated with CPP-P1 showed significant increases in SYN1 in both the IPL (*p* = 0.03) and the GCL (*p* = 0.002) compared to the vehicle-treated group (Figure 5).

### 3.7. Mitochondrial SOD2-Level Evaluation

SOD2 is a key mitochondrial enzyme that protects cells from oxidative stress by converting superoxide radicals into hydrogen peroxide and oxygen. In the retina, where high metabolic activity increases vulnerability to oxidative damage, assessing SOD2 expression helps reveal the mitochondrial response to stress and the potential antioxidant benefits of CPP-P1 in preserving retinal health under elevated intraocular pressure. SOD2 expression was measured by integrated density calculated in the IPL and the GCL. No significant differences were observed across all groups in the expression of SOD2 in the GCL. However, there was a trend toward increased SOD2 expression in the CPP-P1-treated group (*p* = 0.16) (Appendix A).

### 3.8. Assessment of RGC Survival via RBPMS-Positive Cell Analysis

To comprehensively evaluate the neuroprotective effects of CPP-P1 following SO removal, we analyzed RGC survival using RBPMS-positive cell analysis. We monitored PERG responses, integrating both structural and functional assessments to understand the retinal preservation under SO-induced stress. In this round of experiments, SO was removed from the eyes, and PERG was monitored for an additional 3 weeks before tissue collection. Following SO injection and subsequent removal, a significant decrease in PERG amplitude was observed (*p* < 0.0001). This reduction was consistent across rats treated with either vehicle or CPP-P1, showing minimal difference between the two groups (Figure 6A). However, analysis of PERG latency revealed that retinas treated with CPP-P1 exhibited significant preservation compared to those treated with the vehicle (Figure 6B). Images were taken from each of the four quadrants of the retina, and the OD (right) eyes were used as a contralateral control. The average number of RBPMS-positive cells was 1445 ± 45, 831 ± 209, and 1366 ± 154 per mm^2^ for the OD eyes, the SO + vehicle OS (left) eyes, and the SO + CPP-P1 OS eyes, respectively (Figure 6C). As a result, the SO-induced IOP elevation showed an approximately 42% loss in RGCs (*p* = 0.009), while the addition of CPP-P1 reduced this loss by about 37% (*p* = 0.03).

## 4. Discussion

The SO model of ocular hypertension effectively induced retinal damage, as evidenced by the significant loss of RGCs (Figure 1) and marked thinning of the ONL, INL, IPL, and GCL layers (Figure 2). Rats that received an intravitreal injection of CPP-P1 displayed promising indications of neuroprotection, consistent with previous studies showing its ability to preserve RGCs in other glaucoma models [22,26]. Examination of the retinal flat mounts labeled with RGC marker RBPMS revealed that the pattern of RGC degeneration induced by SO was unevenly distributed across the retina (Figure 6). This distinct pattern of degeneration was consistent with acute retinal arterial ischemia [52], possibly due to the high extent of IOP elevation used in this model. Interestingly, RGCs situated near major blood vessels demonstrated better survival, potentially due to improved oxygen supply and nutrient delivery in these areas. This observation suggests that CPP-P1 treatment may enhance the ability of RGCs to withstand stress by leveraging the proximity to vascular support, thereby promoting neuroprotection under conditions of elevated IOP.

After confirming the overall effectiveness of the SO model and the neuroprotective effects of CPP-P1, we shifted our focus to analyzing the potential mechanisms of action of CPP-P1 by evaluating the relative expression of key proteins involved in neuroprotection.

CREB signaling has emerged as a cornerstone in mediating neuroprotective effects in various neurodegenerative models. Activation of CREB can upregulate essential genes such as BDNF, BCL-2, and COX-2, which collectively contribute to cellular resilience under stress. This study underscores the significance of p-CREB as a pivotal mediator of the neuroprotective effects of CPP-P1, further validated by consistent trends observed across similar glaucoma models. CREB-mediated activation of neuroplasticity and neuroprotection have implications for the development of neuroprotective strategies for numerous neurodegenerative disorders, including Alzheimer’s disease, Huntington’s disease, amyotrophic lateral sclerosis, and Parkinson’s disease [53]. Our previous study suggested that CREB signaling is activated following CPP-P1 administration, which is supported by RNA expression of primary RGCs and human retinal tissue and by immunohistochemistry of primary RGCs [30]; therefore, the first two targets analyzed in this study were CREB and its phosphorylated form, p-CREB (Ser133). While CREB is known to promote cell survival and respond to growth factors, some markers of synaptogenesis were also implicated in CPP-P1’s possible mode of neuroprotection [54,55,56]. Therefore, two additional markers were selected: GAP43 and SYN1. These markers were both found to be increased by CPP-P1 in the RNA sequencing data from our previous study (Appendix A) [30]. Additionally, we chose another target from that previous transcriptomic study (Appendix A) [30], SOD2, which may further support CPP-P1’s role in promoting mitochondrial health. Fluorescent microscopy was utilized using consistent settings for laser intensity, exposure time, and gain across all samples to accurately assess the relative expression of these targets. When evaluating the expression of CREB and p-CREB, the naïve group showed a low constitutive expression of CREB and a much higher (approximately two-fold) expression of p-CREB (Figure 3). The ratios of activated p-CREB to CREB were consistent between the SO groups treated with CPP-P1 or vehicle. However, when examining the individual expression levels, the CPP-P1-treated group showed higher overall expressions of both CREB and p-CREB. This suggests that while CPP-P1 treatment did not appear to alter the activation ratio of p-CREB to CREB significantly, it did lead to an overall increase in the expression of both CREB and p-CREB in this acute hypertension model. Synaptogenesis markers GAP43 (Figure 4) and SYN1 (Figure 5) were assessed in the GCL and IPL. SYN1 binds synaptic vesicles and was primarily expressed in the IPL (where the axons of bipolar cells synapse with the dendrites of the RGCs) and the NFL—the RGC axons. A thinning of these layers was found based on the data of H&E staining and significant decreases in SYN1 expression in the SO and vehicle-treated group compared to the naïve group. GAP43 expression was similarly low in both the naïve and SO + vehicle-treated groups, and there was an increasing trend in IPL and GCL in the SO + CPP-P1-treated groups. This observation suggested a modest increase in synaptogenesis or, more likely, the upregulation of proteins that support axonal health and maintenance. These proteins may promote synaptic connectivity, stabilize neural networks, or enhance axons’ structural integrity, contributing to neuronal survival and function. It is worth noting that not all data in experimental biology need to be statistically significant to be biologically meaningful. Trends observed in GAP43 and SYN1 expression, although not statistically robust, align with the hypothesis of enhanced synaptogenesis and axonal stabilization. These findings suggest a potential therapeutic role that warrants further investigation in larger cohorts.

Our use of both structural (retinal-layer thickness) and functional (PERG and RBPMS counts) endpoints provides a comprehensive assessment of CPP-P1 efficacy. This multimodal approach strengthens the reliability of the results, as similar methods have been validated in prior studies of retinal neurodegeneration.

SOD2 is a key enzyme that provides protection against oxidative damage mediated by superoxides generated as a byproduct of mitochondrial bioenergetics and oxidative phosphorylation. The data presented here indicate that CPP-P1 had little to no effect on the SOD2 expression in the SO model (Appendix A). In contrast, transcriptomic data obtained in a chronic ocular hypertension model [30] (Appendix A) suggested that more SOD2 was produced in the IOP-elevated group treated with CPP-P1 compared to the untreated IOP-elevated group. The data shown here suggest there was no increase in the protein expression level, and thus, there was no direct association between CPP-P1 treatment and SOD2 levels in our acute SO model. Investigating the expression at later time points, 2 to 3 weeks of IOP elevation with weekly drug treatment, may show higher SOD2 levels upon CPP-P1 treatment. Some studies suggest that SOD2 is upregulated early in the disease processes as a compensatory mechanism, gradually decreasing over time [57,58]. Therefore, it is possible that CPP-P1 may help to sustain the elevated SOD2 expression during this period.

The variability in responses, such as SOD2 expression, reflects the inherent complexity of biological systems. Such variations are not necessarily indicative of experimental limitations but rather underline the importance of studying temporal and context-specific dynamics in oxidative stress pathways.

The novel use of CPP-P1 in the SO model demonstrated its ability to mitigate severe retinal damage, offering a therapeutic avenue for conditions like acute angle-closure glaucoma. This work lays a foundation for exploring long-term applications and further mechanistic insights, including downstream targets of CREB signaling. It is also important to acknowledge the limitations of this study. The IOP measurements were carried out consistently in our hands; however, there has been evidence of anesthesia (e.g., isoflurane) influencing IOP values [59,60]. The consistency of injury in the SO model of ocular hypertension has not yet been well characterized, and the possible variability makes the assessment of CPP-P1’s efficacy difficult, especially with the model being acute. Regarding the inability to accurately assess PERG when SO was present in the anterior chamber of experimental animals, the group that developed this model in mice had similar issues, noting that the “lack of progression of PERG amplitude reduction suggests SO itself may affect the light stimulation and PERG signal” [48]. While the inability to measure PERG during the presence of SO posed a limitation, the subsequent removal and validation through RBPMS-positive counts provided reliable indicators of retinal health. These methodological adaptations ensure that the observed effects of CPP-P1 are robust and interpretable. Another potential limitation to consider is the form of CPP-P1 used, and a more stable mode of delivery that allows for sustained long-term release would be desirable. There are many other targets of the CREB transcription factor to be investigated in the future, such as BCL-2, BDNF, Leptin, Trkb, and COX2, to name a few [61].

Finally, while this study provides valuable insights into the expression patterns of key proteins through immunohistochemistry, a secondary confirmation using complementary techniques, such as Western blotting or quantitative proteomics, would strengthen the findings. Unfortunately, these analyses were not conducted due to constraints, including limited tissue availability, highlighting the need for future studies to address this limitation and provide more comprehensive validation of the observed protein expression changes. The findings presented in this work align with previous studies demonstrating that neuroprotective agents targeting CREB pathways can prevent RGC degeneration under stress conditions.

## 5. Conclusions

In this study, an acute SO model of glaucoma was used to provide additional evidence of the multifaceted neuroprotective nature of CPP-P1 and to investigate the cellular and signaling mechanisms of action of peptain-1. The immunohistochemical staining demonstrated an increase in overall CREB expression as well as p-CREB expression in rats treated with CPP-P1, indicating the involvement of this critical signaling pathway in CPP-P1’s neuroprotective effects, which include the preservation of cell counts and retinal thickness. There were trends of increased synaptogenesis (indicated by GAP43 and SYN1) and mitochondrial antioxidant response, as indicated by SOD2. Other targets downstream of CREB may be more favored in this context and will be investigated in future studies. The retinal injury in this model was too acute to precisely assess cell survival and preservation of function. This study provides additional evidence supporting CPP-P1’s potential to be developed into a neuroprotective agent for the treatment of glaucomatous optic neuropathy as well as more acute neurodegenerative diseases, such as acute angle-closure glaucoma or central retinal artery occlusion.

## Figures and Tables

**Figure 1 biomolecules-15-00219-f001:**
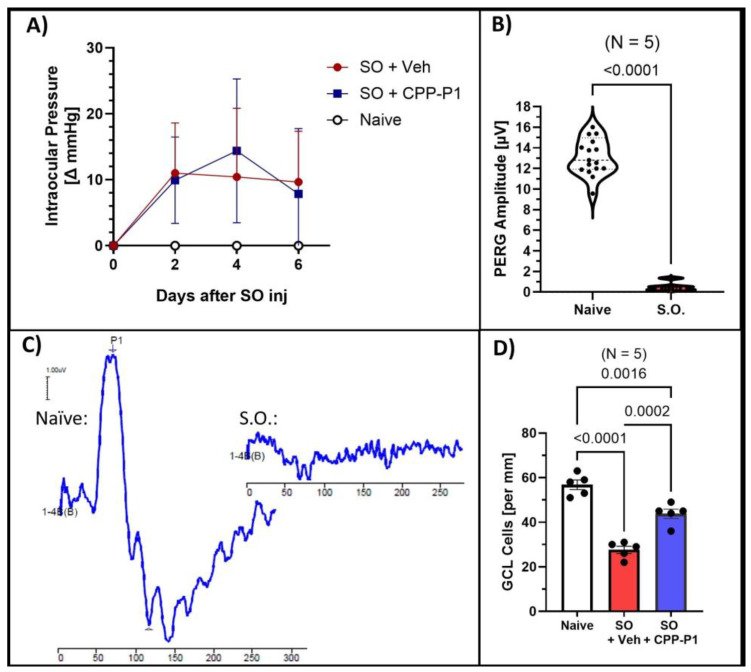
Damage produced by silicone oil (SO)-induced ocular hypertension and CPP-P1-mediated RGC protection. (**A**) Intraocular pressure (IOP) measurements indicated a significant (*p* < 0.01) elevation in IOP occurred following SO injection. Delta mmHg was calculated compared to respective baseline values. (**B**) Pattern electroretinogram (PERG) amplitude measurements of healthy eyes (naïve) compared to those injected with SO. (**C**) Representative PERG graphs were obtained from the eyes with and without SO. (**D**) GCL counts from retinal cross-sections showed significant preservation of GCL cells with CPP-P1 treatment.

**Figure 2 biomolecules-15-00219-f002:**
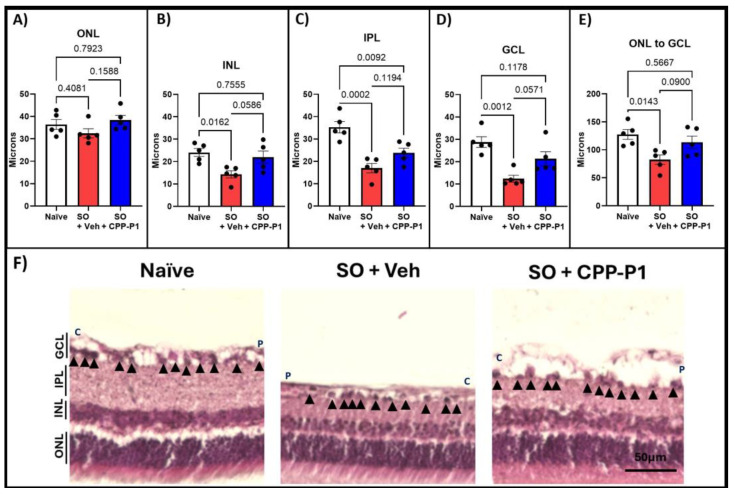
Retinal-layer thickness measurements following H&E staining. Thickness measurements were taken from the (**A**) outer nuclear layer (ONL), (**B**) inner nuclear layer (INL), (**C**) inner plexiform layer (IPL), (**D**) ganglion cell layer (GCL), and (**E**) total retinal thickness. (**F**) Representative images illustrate the significant loss in retinal thickness following SO injection and the preservation of retinal layers with CPP-P1 administration. Black arrowheads indicate the cells in the RGC layer that were analyzed and included in Figure 1D. C indicates the more central location and P indicates the more peripheral location. Scale bar 50 µm.

**Figure 3 biomolecules-15-00219-f003:**
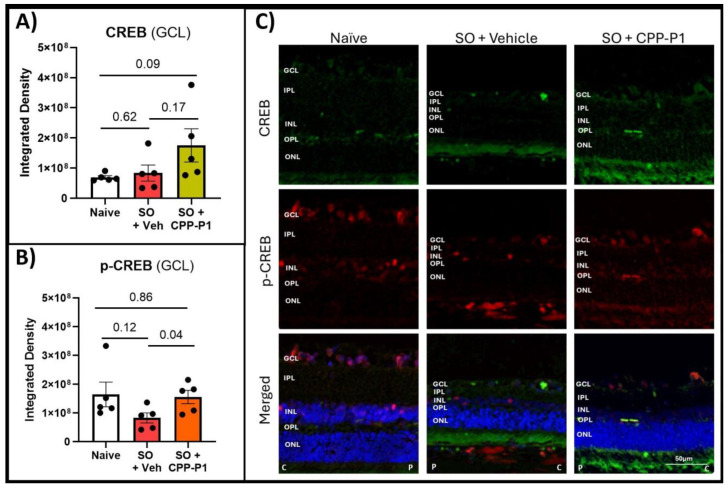
CREB and p-CREB levels in the experimental retinas. (**A**) CREB fluorescence in GCL, (**B**) p-CREB fluorescence in GCL. p-CREB levels were significantly elevated with CPP-P1 treatment compared to the vehicle-treated group. (**C**) Representative images, CREB (green), p-CREB (red), DAPI (blue). C represents the direction toward the central region, while P indicates the direction toward the peripheral region. Scale bar 50 µm.

**Figure 4 biomolecules-15-00219-f004:**
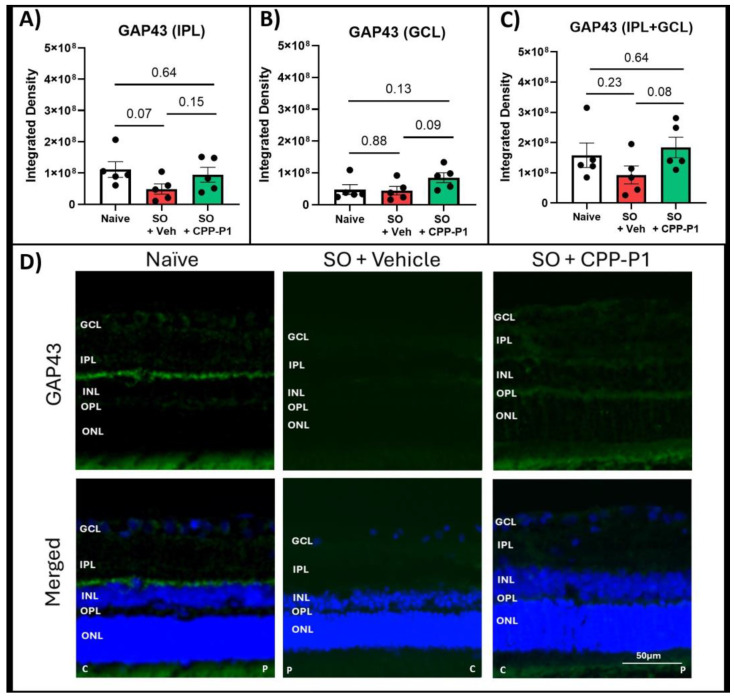
Expression of synaptogenesis marker GAP43. GAP43 expression levels are shown in the (**A**) inner plexiform layer, (**B**) ganglion cell layer, and (**C**) combined inner plexiform layer and ganglion cell layer, with (**D**) representative images. GAP43 (green), DAPI (blue). C: central orientation, P: peripheral orientation. Scale bar 50 µm.

**Figure 5 biomolecules-15-00219-f005:**
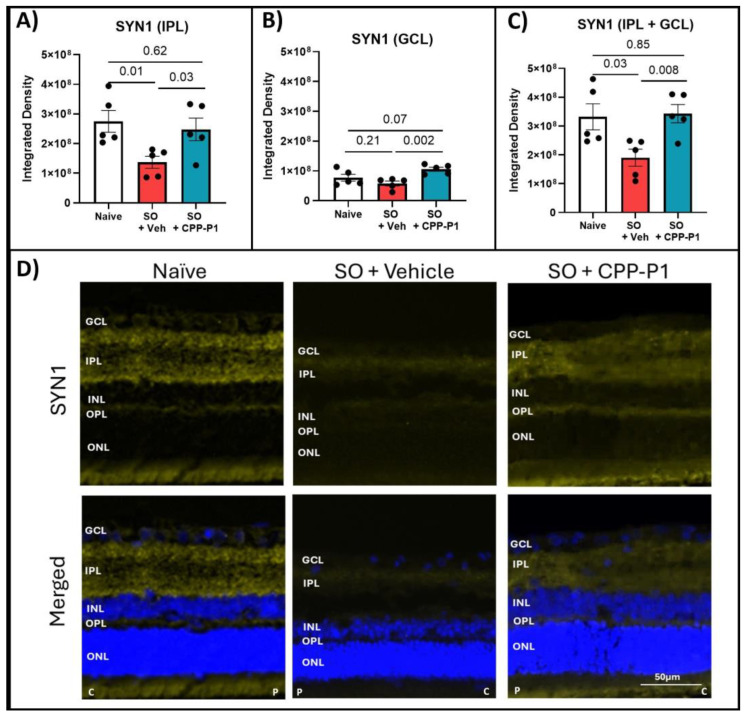
Expression of synaptogenesis marker SYN1 in the IPL and GCL. (**A**) Inner plexiform-layer measurements, (**B**) ganglion cell layer, (**C**) combined inner plexiform layer and ganglion cell layer, (**D**) representative images. SYN1 expression was significantly increased in both the IPL and the GCL following CPP-P1 treatment. SYN1 (yellow), DAPI (blue). C indicates the more central location and P indicates the more peripheral location. Scale bar 50 µm.

**Figure 6 biomolecules-15-00219-f006:**
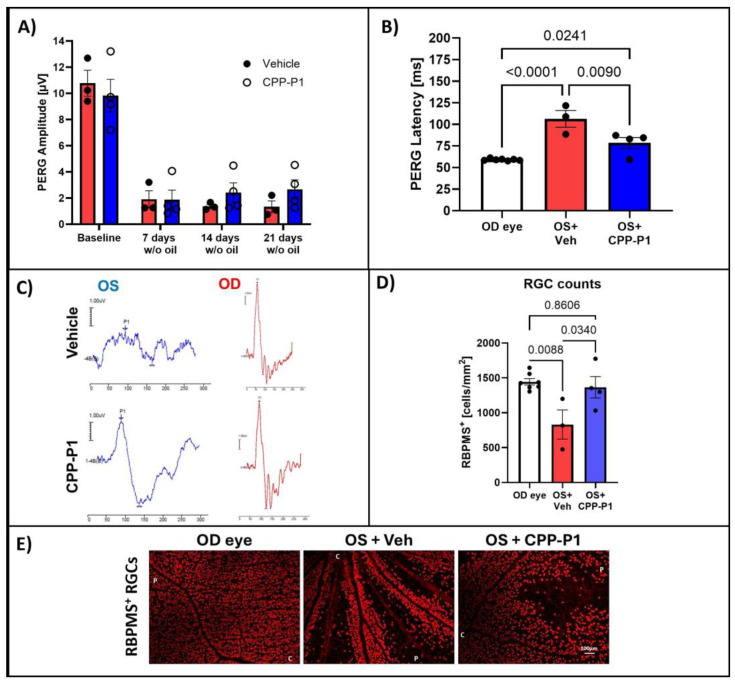
PERG measurements and RBPMS-positive RGC counts following SO removal. (**A**) PERG amplitude measurements in OS eyes were recorded before SO injection (baseline) and weekly for 3 weeks after SO removal (w/o oil). (**B**) PERG latency averaged across all post-removal measurements showed a significant reduction in the response time delay of RGCs to a patterned visual stimulus. (**C**) Representative PERG graphs following SO removal. (**D**) RGC counts were obtained from RBPMS-stained retinal flat mounts. (**E**) Representative images of RBPMS-stained retinas. N = 3–4 animals; statistical analysis was performed using one-way ANOVA. Images were captured at 10× magnification. C marks the central direction, while P marks the peripheral direction. The scale bar represents 100 µm.

**Table 1 biomolecules-15-00219-t001:** Quantification of retinal-layer thickness following H&E staining. * *p* < 0.05, ** *p* < 0.01, *** *p* < 0.001.

Group	% Loss in Vehicle	*p*-Value Vehicle: Naïve	% Loss in CPP-P1	*p*-Value CPP-P1: Vehicle	*p*-Value CPP-P1: Naïve
ONL	11%	0.408	0%	0.159	0.792
INL	41%	* 0.016	9%	0.059	0.756
IPL	52%	*** 0.0002	33%	0.119	** 0.009
GCL	57%	*** 0.001	26%	0.057	0.118
Total	65%	* 0.014	11%	0.090	0.567

**Table 2 biomolecules-15-00219-t002:** Analysis of CREB expression levels in the experimental retinas. The table represents the relative CREB and phosphorylated CREB (p-CREB) levels in each experimental group, expressed in arbitrary units. Additionally, the ratio of p-CREB to total CREB is provided. Statistical significance is indicated by * *p* < 0.05.

	Naïve	*p*-Value Vehicle: Naïve	Vehicle	*p*-Value CPP-P1: Vehicle	CPP-P1	*p*-Value CPP-P1: Naive
CREB	6.9E7	0.62	8.3E7	0.17	1.8E8	0.09
p-CREB	1.6E8	0.12	8.3E7	* 0.04	1.6E8	0.86
p-CREB/CREB	2.3		1		0.9	

## Data Availability

The datasets generated and analyzed during the current study are available from the corresponding author upon reasonable request.

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
