# Peer review of "A Neuroprotective Peptide Modulates Retinal cAMP Response Element-Binding Protein (CREB), Synapsin I (SYN1), and Growth-Associated Protein 43 (GAP43) in Rats with Silicone Oil-Induced Ocular Hypertension"

_biomolecules, 2025, doi:10.3390/biom15020219_

Round 1
Reviewer 1 Report
Comments and Suggestions for Authors
The manuscript used silicon oil elevated IOP in rats as an animal model of glaucoma to study the mechanism of action of a neural protective peptide CPP-P1 which was previously published by the authors’ group that showed protective effects to multiple animal models of glaucoma. The authors used SO injection to the anterior chamber of the rat eye to elevate IOP, then use intravitreal injection to deliver CPP-P1 peptide or PBS vehicle to the posterior chamber of the eye. They used retinal histology of retinal cross sections, and retinal flat mount immunostaining of RGC marker to count RGC cell number and evaluate the effect of IOP elevation and CPP-P1 treatment to retinal morphology. They also used Pattern ERG to measure RGC function. Through IHC, they showed CREB staining and phosphorylated CREB staining both elevated by CPP-P1 treatment, and the authors also claimed that two CREB target genes also shown to be increased by IHC, including synapsin 1 and GAP43. The authors also immunostained an antioxidant protein SOD2 and did not observe significant increase by CPP-P1 in this model. They concluded that the neural protection by CPP-P1 injection in this glaucoma model is via the CREB signaling pathway.
While there are not many neural protective agents available for RGC survival and function in glaucoma treatment, this study as a continuation of years of work on CPP-P1 in translational development of RGC preserving therapy for glaucoma patients is important. However, there are several issues to be addressed.
1. About the model: to generate elevated IOP, one injection of SO was performed, and then the CPP-P1 treatment is through another intravitreal injection. The authors need to provide more information about whether both SO and peptide injections were performed at the same time and whether the intravitreal injection would affect IOP because a hole was made to the posterior chamber.
2. How would CPP-P1 pass through retinal cell layers to protect INL, in addition to RGC? Is there any immunostaining of the peptide to show where it is distributed after the injection?
3. Many of the critical data are not statistically significant, including retinal thickness, PERG responses, CREB level and so on, suggesting either the sample size is not sufficient to address the authors’ hypothesis, or the method is not sensitive enough to detect the small changes. Other than IHC, there should be alternative methods, such as qPCR and western blots to validate the critical point about the CREB and P-CREB levels. If SYN1 and GAP34 are CREB target genes, qPCR should be the first experiment to be performed.
4. For the MOA of CPP-P1, it is still unclear how a short peptide, after getting into the cells, which target does it binds to, and how CREB expression and phosphorylation been affected. Perhaps some cell based assay using primary neurons with hypoxia challenge wi/o CPP-P1 can help identify the direct target of this neuroprotective peptide.
5. For immunohistochemistry, the orientation of the eyes should be marked before sections are made and the area of the images should be labeled because the RGC distribution is not even from centro to peripheral and from superior to inferior, temporal to nasal. Same applies to the retinal flat mounts.
6. The averaged flash response of PERG from CPP-P1 treated animals should be shown in Fig 6.
7. For PERG responses and RGC cell numbers are not coherent. The vehicle control showed 50% RGC cell loss compared to Naïve control, while the PERG signals were completely ablated. Does this mean the rest RGCs are not well functioning as well, or the PERG is not sensitive to detect the left over 50% RGC cell function? How much of such reduction is due to SO blurring the PERG signaling vs dampened RGC function?
8. Minor: in Method for IHC of retinal cross section and flat mount immunostaining, secondary antibody should be specified for their species specificity, conjugation and vendor information.
Author Response
Dear Editors of Biomolecules,
We would like to express our sincere gratitude to the reviewers for their insightful comments and constructive feedback on our manuscript titled "A Neuroprotective Peptide Modulates Retinal cAMP Response Element-Binding Protein (CREB), Synapsin I (SYN1), and Growth-Associated Protein 43 (GAP43) in Rats with Silicone Oil-Induced Ocular Hypertension". We appreciate the time and effort invested in reviewing our work, which has undoubtedly enhanced the quality and rigor of our research. In response to the reviewers' comments, we have carefully revised the manuscript and have provided detailed responses and explanations for the changes made. We believe that these revisions enhance clarity, and strengthen the validity and impact of our findings. Below, we outline our responses to each reviewer's comments and describe the corresponding modifications made to the manuscript.
Reviewer 1
- About the model: to generate elevated IOP, one injection of SO was performed, and then the CPP-P1 treatment is through another intravitreal injection. The authors need to provide more information about whether both SO and peptide injections were performed at the same time and whether the intravitreal injection would affect IOP because a hole was made to the posterior chamber.
Response:
We appreciate the reviewer’s question regarding the timing and potential impact of intravitreal injections on intraocular pressure (IOP) stability.
To clarify, the CPP-P1 injections were administered immediately following the silicone oil (SO) injection. This approach ensured that the SO-induced IOP elevation was established prior to peptide administration. While an intravitreal injection can cause a transient fluctuation in IOP, this effect is generally short-lived and typically resolves within a few hours as the injection site seals and ocular pressure stabilizes. Importantly, any potential impact on IOP was minimal and did not interfere with the sustained IOP elevation induced by the SO. Additionally, past studies using this model have shown that such injection procedures do not significantly impact IOP in the long term, as the site of injection in the posterior chamber closes and fully heals within a few days.
- How would CPP-P1 pass through retinal cell layers to protect INL, in addition to RGC? Is there any immunostaining of the peptide to show where it is distributed after the injection?
Response:
We appreciate the reviewer’s question regarding the potential penetration and distribution of CPP-P1 within the retinal layers, particularly its ability to reach the inner nuclear layer (INL) in addition to the retinal ganglion cells (RGCs).
While we did not perform immunostaining to track CPP-P1 distribution specifically in this study, previous research using a Cy7-labeled analog of the peptide (Cy7-peptain-1) demonstrated its presence in multiple retinal layers, including, the ganglion cell layer (GCL), inner plexiform layer (IPL), outer plexiform layer (OPL), and the inner and outer segments of photoreceptors—following systemic administration [Stankowska et al. 2019, Cell Death Discovery]. This distribution pattern suggests that CPP-P1 can potentially penetrate beyond the RGC layer.
With the more localized intravitreal delivery of CPP-P1 in our study, it is likely that the higher concentration achieved in the vitreous could facilitate even deeper penetration into retinal layers, reaching the INL. Additionally, after initial uptake by RGCs, it is possible that CPP-P1 may undergo intercellular transfer, potentially extending its protective effects to adjacent cells within the INL.
Future studies employing direct immunostaining of CPP-P1 could provide further insights into its retinal distribution and verify its localization within specific cell layers post-injection.
Stankowska, D.L., Nam, M. H., Nahomi, R. B., Chaphalkar, R. M., Nandi, S. K., Fudala, R., Krishnamoorthy, R. R., & Nagaraj, R. H., Systemically administered peptain-1 inhibits retinal ganglion cell death in animal models: Implications for neuroprotection in glaucoma. . Cell Death Discovery, 2019. 5.
- Many of the critical data are not statistically significant, including retinal thickness, PERG responses, CREB level and so on, suggesting either the sample size is not sufficient to address the authors’ hypothesis, or the method is not sensitive enough to detect the small changes. Other than IHC, there should be alternative methods, such as qPCR and western blots to validate the critical point about the CREB and P-CREB levels. If SYN1 and GAP34 are CREB target genes, qPCR should be the first experiment to be performed.
Response:
Thank you for the valuable feedback on the statistical significance of the findings. We agree that, due to the limited sample size, some results—such as retinal thickness, PERG responses, and CREB levels—did not reach statistical significance. However, we believe these data still provide meaningful insights into trends that align with our hypothesis and may suggest a protective role of CPP-P1 in retinal health.
While a larger sample size would undoubtedly increase the power to generate statistically significant data, the trends observed here are consistent with our expectations based on previous studies and point towards a potentially beneficial modulation of retinal integrity and signaling pathways, including CREB and p-CREB.
Regarding validation of critical markers like CREB, we opted to focus on protein-level assessments due to tissue limitations, prioritizing immunohistochemistry over qPCR or Western blotting. As previously published data included qPCR on CREB expression, this study aimed to extend those findings by examining CREB at the protein level. Despite the lack of statistical significance across all measures, these findings collectively suggest trends that would warrant further investigation in a larger study.
In future work, employing both qPCR and Western blotting alongside immunohistochemistry with a larger cohort could help confirm the transcriptional regulation of CREB targets such as SYN1 and GAP34 and provide a more comprehensive understanding of CPP-P1’s effects on retinal protection.
- For the MOA (mechanism of action) of CPP-P1, it is still unclear how a short peptide, after getting into the cells, which target does it binds to, and how CREB expression and phosphorylation been affected. Perhaps some cell based assay using primary neurons with hypoxia challenge wi/o CPP-P1 can help identify the direct target of this neuroprotective peptide.
Response:
Thank you for these valuable insights. We agree that further elucidating the mechanism of action (MOA) of CPP-P1 is crucial to fully understanding its neuroprotective potential. Future studies will indeed aim to investigate the direct intracellular targets of CPP-P1 and clarify how it influences CREB expression and phosphorylation.
Given the multifunctional nature of the parent protein, αB-crystallin, it is plausible that CPP-P1 may engage multiple cellular mechanisms to achieve neuroprotection. This complexity suggests that CPP-P1 may not act on a single target but rather modulates a network of protective pathways within the cell. A cell-based assay using primary neurons under hypoxia, with and without CPP-P1, is an excellent suggestion to help identify direct targets of the peptide and better understand its interaction within stressed neuronal environments.
Our previously published work has demonstrated that CPP-P1 effectively protects RGCs from NF deprivation and ET-1 mediated cell death, underscoring its potential in neuroprotective applications.
As highlighted in Dr. Shigemi Matsuyama’s research, the mechanisms behind cell penetration by CPPs, including CPP-P1, are not yet fully understood. The positively charged amino acids in certain CPP sequences, like TAT and poly-arginine, facilitate interactions with negatively charged cell surface components, such as proteoglycans or phospholipids, potentially initiating endocytosis or macropinocytosis. However, studies also indicate that some cell-penetrating peptides (CPP5s), like VPTLK and KLPVM, may bypass traditional endocytosis and pinocytosis pathways. CPP5s have been observed to enter cells even in the presence of endocytosis inhibitors, suggesting alternative mechanisms at play.
Mass spectrometry studies have shown that CPP5 peptides can reach high intracellular concentrations, with minimal cytotoxicity, making them viable tools for cellular delivery. For example, VPTLK and KLPVM have demonstrated the ability to transduce Cre protein, indicating a potential for drug delivery applications without significant cellular toxicity.
Future research on CPP-P1 could build on these findings, exploring the specific amino acid composition and structural attributes that enable cell penetration. This approach could shed light on the peptide’s intracellular pathways and identify specific protein interactions involved in modulating CREB and P-CREB levels.
Gomez, J.A., et al., Cell-Penetrating Penta-Peptides (CPP5s): Measurement of Cell Entry and Protein-Transduction Activity. Pharmaceuticals (Basel), 2010. 3(12): p. 3594-3613.
- For immunohistochemistry, the orientation of the eyes should be marked before sections are made and the area of the images should be labeled because the RGC distribution is not even from centro to peripheral and from superior to inferior, temporal to nasal. Same applies to the retinal flat mounts.
Response:
Thank you for this valuable observation. We agree that marking the orientation of the eyes prior to sectioning would have improved our ability to specify retinal regions in the images, considering the non-uniform distribution of RGCs across central to peripheral, superior to inferior, and temporal to nasal axes. While we have updated the figures to reflect the central to peripheral distribution, unfortunately, due to the lack of initial orientation marking upon eye removal, we are unable to provide information on the other axes in this study.
In future work, we will incorporate orientation markings to ensure more precise localization and labeling in both sectioned images and retinal flat mounts. This will allow for a more accurate assessment of RGC distribution across all axes.
- The averaged flash response of PERG from CPP-P1 treated animals should be shown in Fig 6.
Response:
The figure has been updated to include representative PERG graphs.
- For PERG responses and RGC cell numbers are not coherent. The vehicle control showed 50% RGC cell loss compared to Naïve control, while the PERG signals were completely ablated. Does this mean the rest RGCs are not well functioning as well, or the PERG is not sensitive to detect the left over 50% RGC cell function? How much of such reduction is due to SO blurring the PERG signaling vs dampened RGC function?
Response:
We appreciate the insightful observations regarding the PERG and RGC cell count findings in our study.
To clarify, the complete ablation of the PERG signal in the vehicle control group, despite a 50% reduction in RGC numbers, suggests that the remaining RGCs may not be fully functional. This supports the hypothesis that the surviving RGCs might be compromised in their electrophysiological response capability, even if they retain cellular integrity. Alternatively, it raises the possibility that the PERG may not be sufficiently sensitive to detect the diminished functional capacity of these remaining RGCs.
Additionally, we recognize that silicone oil (SO) may interfere with the accurate recording of PERG signals by contributing to light diffusion or blurring, which could dampen the signal independently of actual RGC function. Previous research in similar animal models has also indicated that SO can affect the PERG signal independently, potentially attenuating the amplitude of the response due to its impact on light transmission. As noted in the results, this limitation was acknowledged, and our findings align with previous observations that SO’s presence in the anterior chamber may contribute to reduced sensitivity in detecting subtle functional changes in surviving RGCs.
Moving forward, we acknowledge that a more stable and sustained delivery method for CPP-P1 could enhance the potential for observing longer-term functional preservation of RGCs.
- Minor: in Method for IHC of retinal cross section and flat mount immunostaining, secondary antibody should be specified for their species specificity, conjugation and vendor information.
Response:
IHC methods updated to include secondary antibody information.
Reviewer 2 Report
Comments and Suggestions for Authors
please find enclosed

Author Response
Dear Editors of Biomolecules,
We would like to express our sincere gratitude to the reviewers for their insightful comments and constructive feedback on our manuscript titled "A Neuroprotective Peptide Modulates Retinal cAMP Response Element-Binding Protein (CREB), Synapsin I (SYN1), and Growth-Associated Protein 43 (GAP43) in Rats with Silicone Oil-Induced Ocular Hypertension". We appreciate the time and effort invested in reviewing our work, which has undoubtedly enhanced the quality and rigor of our research. In response to the reviewers' comments, we have carefully revised the manuscript and have provided detailed responses and explanations for the changes made. We believe that these revisions enhance clarity, and strengthen the validity and impact of our findings. Below, we outline our responses to each reviewer's comments and describe the corresponding modifications made to the manuscript.
Reviewer 2
How did the animals' behavior change when the eye pressure was elevated and maintained at the 30Hgmm, was an analgesia necessary?
Response:
No change in behavior was noted, and thus, no analgesia was used. Only one eye was injected out of concern for animal welfare.
We remain committed to advancing scientific knowledge in the field of neuroprotection and are grateful for the opportunity to contribute to the discourse in Biomolecules.
Round 2
Reviewer 1 Report
Comments and Suggestions for Authors
The authors addressed most of my comments. The revised manuscript showed stronger scientific rigor.